# Crohn’s Disease and Female Infertility: Can Nutrition Play a Supporting Role?

**DOI:** 10.3390/nu14122423

**Published:** 2022-06-10

**Authors:** Alessandra Vincenti, Federica Loperfido, Rachele De Giuseppe, Matteo Manuelli, David Bosoni, Alessandra Righi, Rossella E. Nappi, Hellas Cena

**Affiliations:** 1Laboratory of Dietetics and Clinical Nutrition, Department of Public Health, Experimental and Forensic Medicine, University of Pavia, 27100 Pavia, Italy; alessandra.vincenti01@universitadipavia.it (A.V.); rachele.degiuseppe@unipv.it (R.D.G.); hellas.cena@unipv.it (H.C.); 2Clinical Nutrition and Dietetics Service, Unit of Internal Medicine and Endocrinology, ICS Maugeri IRCCS, 27100 Pavia, Italy; md.matteomanuelli@gmail.com; 3Department of Clinical, Surgical, Diagnostic and Pediatric Sciences, University of Pavia, 27100 Pavia, Italy; david.bosoni01@universitadipavia.it (D.B.); alessandra.righi01@universitadipavia.it (A.R.); rossella.nappi@unipv.it (R.E.N.); 4Research Center for Reproductive Medicine, Gynecological Endocrinology and Menopause, IRCCS San Matteo Foundation, 27100 Pavia, Italy

**Keywords:** IBD, inflammatory bowel diseases, Crohn’s disease, fertility, malnutrition, nutrition

## Abstract

Crohn’s disease (CD) is a chronic inflammatory disease (IBD) that can affect the entire gastrointestinal tract in a non-continuous mode. CD is generally diagnosed most commonly between 15 and 35 years of age and may affect female fertility. The role of diet in supporting wellbeing outcome and reproductive potential in women is well-known; however, no effective efforts have been made to improve women’s awareness in CD. Our review aims to describe the burden of CD on women’s fertility, reporting the most relevant nutrients that support reproductive function to ensure women diagnosed with IBD an adequate health-related quality of life.

## 1. Introduction

Inflammatory bowel diseases (IBDs), including Crohn’s disease (CD) and ulcerative colitis (UC), are gastrointestinal bowel-inflammatory disorders that differently affect the intestine [1]. Specifically, CD affects the entire gastrointestinal tract in a non-continuous mode, most commonly the ileum or perianal region [2].

CD evolution is typically characterized by periods of remission with the absence of symptoms, interspersed between flares of intestinal inflammation [1]. During active phases, common symptoms mainly include diarrhea and abdominal pain [1]. Other symptoms such as weight loss, fatigue, fever, vomiting, and nausea, as well as related symptoms due to recurrent fistulas or other perianal issues (ulcers or fissures) may also be present [1].

All those symptoms might contribute to impairing nutritional status, thus leading to malnutrition. Indeed, the reported prevalence of malnutrition in patients with CD ranges between 20% and 85% [3] depending on the activity, duration, location, extent of disease, and magnitude of the systemic inflammatory response [1]. 

A decrease in food intake, as well as a restricted diet, characterized by selective and avoiding behavior towards the food items perceived as linked to the worsening of symptoms (e.g., dairy products and fiber-rich foods), may lead to nutritional inadequacies or even deficiency [4]. These include mineral deficiencies (e.g., iron, calcium, zinc, selenium), folate, Vitamin B12, Vitamin D, Vitamin A, Vitamin K and/or the inadequate intake of proteins and fats [5,6,7]. Nutritional deficiency may also be a consequence of nutrient loss due to both mucosal alteration (e.g., impaired epithelial transport and the loss of epithelial integrity) [5] and small intestinal bacterial overgrowth (SIBO). In the latter case, increased intestinal permeability and gastrointestinal transit acceleration reduce the contact time of the luminal content with the mucosal surface and therefore lead to malabsorption [8]. 

On the other hand, medications often induce nausea, vomiting, and anorexia, or interfere with micronutrient absorption including phosphorus, zinc, and calcium [6], or folic acid as in long-term sulfasalazine therapy with consequent anemia [9]. 

Moreover, surgery itself may induce iatrogenic malabsorption [8]. Indeed, the resection of large, small-bowel segments is usually followed by reduced digestion and the absorption of nutrients and, therefore, by watery diarrhea [10]. Surgical ileum resection of more than 100 cm leads to a negative feedback mechanism, with bile acids returning to the liver exceeding the rate of hepatic synthesis [10]. Because of bile acid pool depletion, fat digestion and absorption are impaired, causing steatorrhea and fat malabsorption [10]. 

Since CD is generally diagnosed most commonly between 15 and 35 years of age, in the middle of the reproductive age [11], fertility can be compromised, even though studies in women have indicated that fertility is normal and infertility rates are low [12] during the inactive phase of the disease. On the other hand, other studies have documented an increased rate of infertility during the active phase, when the nutritional status is likely to be impaired [13,14,15] because of the aforementioned mechanisms. 

Alongside involuntary childlessness, an overall increase in voluntarily choice to avoid pregnancy has been reported [16,17]. A systematic review of eleven studies showed a 17–44% reduction in fertility in women with CD compared to controls, mostly due to voluntary choice rather than physiological causes of infertility [16]. The reasons for voluntary childlessness are multiple (e.g., fear of disease exacerbation, maternal and fetal complications, drug interactions during pregnancy, and fear of child care inability). A brief summary of the common causes of infertility in women with CD is reported in Figure 1. It is worth noting the inverse correlation with IBD-specific pregnancy-related knowledge [18], suggesting that women with CD may benefit from increased knowledge about their illness, including nutritional issues, and its management [18,19].

The role of diet and nutrition education in supporting wellbeing outcome and fertility in women is well-known [20,21]; however, despite evidence, no effective efforts have been made to tailor specific interventions in women affected by CD.

Notwithstanding, this narrative review aims to fill this gap, describing the burden of CD on women’s fertility and focusing on nutritional aspects, as well as on nutrients suitable for fertility protection.

## 2. CD and Fertility: The Supporting Role of Nutrition

Nutritional aspects in women affected by CD are particularly relevant as they potentially influence disease activity and morbidity, as well as fertility.

### 2.1. Micronutrients

In general, in IBD, micronutrients (e.g., Vitamin C, A, E, B vitamins, zinc, selenium, copper, and iron) play a key role in regulating immune response and homeostasis by inhibiting the release of inflammatory cytokines and promoting the differentiation of regulatory (TREG) lymphocytes over pro-inflammatory TH17 lymphocytes [22,23] as well as counteracting reactive oxygen species’ production [22]. ESPEN guidelines recommend that patients with IBD should be checked for micronutrient deficiencies regularly, and specific deficits should be appropriately corrected [5].

Micronutrients deficiencies in CD, as a consequence of appetite loss, inadequate food intake, impaired absorption and gastric losses [6], may also affect fertility through different mechanisms: oocyte quality, maturation, fertilization, and implantation [24,25] for folate; ovulation processes and menstrual cycle alterations for zinc [24]; oocyte quality and blastogenesis for vitamin A [24]; the steroidogenesis of sex hormones (estradiol and progesterone) for vitamin D [26]; and embryogenesis, inflammation, oxidative stress [24] and homocysteine metabolism [27] for B vitamins (i.e., folate, vitamin B6, vitamin B12) and iron.

#### 2.1.1. B Vitamins (Folate, Vitamin B12)

Folate deficiency is very common among women affected by CD [5], with a prevalence of about 28.8% [28]. Deficiency causes are (i) low dietary intake (e.g., the low consumption of high-folate foods including green leafy vegetables, fruits, bread, and fortified cereals), (ii) malabsorption, (iii) drug–nutrient interactions such as dihydrofolate reductase inhibition (methotrexate) or side effects as gastric distress, headache, nausea, vomiting, and anorexia (sulphasalazine) [5]. 

As for folate deficiency, this is the most common cause of total homocysteine increased plasma concentration (hyperhomocysteinemia, HHcy) [29], a risk factor for cardiovascular diseases as well as arterial and venous thromboembolic events observed in CD patients [30]. Additionally, HHcy is strictly related to oxidative stress and promotes the secretion of pro-inflammatory chemokines and the differentiation of CD4 + T cells into pro-inflammatory TH17 cells [31].

HHcY is also involved in the fertility and reproductive process [32], since gametogenesis and early embryogenesis are strongly affected by methylation errors [32].

Moreover, it is well-established that Vitamin B12 deficiency is more common in patients with CD than in those with UC, since a functionally intact ileum is needed to absorb vitamin B12/intrinsic factor complex [27]. Previous studies have reported that CD patients who underwent 60 cm of terminal ileal resection or more eventually developed B12 deficiency [5], highlighting the need to carefully monitor these subsets of patients to avoid clinical complications such as megaloblastic anemia and peripheral neuropathy, HHcy. A thorough systematic review, including 3732 patients from 42 studies, concluded that B12 deficiency in patients affected by CD with no ileal resection (or resection of <30 cm) was no more common than in the general population [33]. 

#### 2.1.2. Vitamin D

The relationship between vitamin D and IBD is likely to be the most outstanding as far as immunity and immune response regulation are concerned [23]. Indeed, vitamin D modulates gut mucosal immunity and intestinal barrier integrity [34] and alters T cell response, favoring TREG cells and inhibiting the production of proinflammatory cytokines [35]. Patients with CD are known to be at higher risk for vitamin D deficiency [36] for several reasons, including (i) intestinal inflammation leading to malabsorption, (ii) bile acid malabsorption, (iii)restricted dietary intake, (iv) reduced sunlight exposure, and (v) immunosuppressive and/or corticosteroid treatment [37].

There is strong evidence correlating vitamin D deficiency with CD activity [38,39,40]. In a retrospective cohort of 131 patients with CD, Ye et al. [40] showed that low or insufficient serum 25-hydroxy vitamin D3 (25(OH)D3) levels (<50 nmol/L and 50–75 nmol/L, respectively) were inversely associated with disease severity when evaluated both by simple endoscopic score for CD (*p* = 0.001), by CD activity index (*p* < 0.001) and C-reactive protein (*p* < 0.001). Again, in a cohort study (*n* = 60 subjects, of which 39 females) with CD or UC, vitamin D serum level was inversely correlated with erythrocyte sedimentation rate and fecal calprotectin concentration (*p* = 0.0001 and *p* = 0.0025, respectively) [39]. Those studies highlighted that low vitamin D serum level was not only associated with clinical symptoms but also with objective markers of inflammation.

Given the observational data linking the association between inadequate vitamin D status and CD, it is not surprising that some researchers have prospectively examined the impact of vitamin D supplementation on CD. However, although several RCTs [41,42,43] involving patients with CD have shown a slight improvement in the clinical parameters after vitamin D supplementation, overall, sufficient evidence has not been provided to support the role of vitamin D on relapse risk decrease [41,43], remission and quality of life [42] because of low statistical significance [43] and/or inadequate sample size [41,42,43]. 

It has been reported that Vitamin D likely participates in the modulation of female reproduction function and fertility, as well as pregnancy and lactation [44,45,46]. This comes from the observation that vitamin D receptors are expressed in numerous tissues of the reproductive organs (e.g., ovaries, endometrium, placenta, pituitary gland, and hypothalamus) [26,46]. Additionally, vitamin D affects various endocrine processes and has a role in steroidogenesis by regulating the expression and the activity of key enzymes involved in this process [46]. 

A systematic review and meta-analysis on vitamin D status, including 11 cohort studies with 2700 women, showed an association between vitamin D status and outcomes of assisted reproductive treatment [47]. Women with adequate vitamin D status (25(OH)D3 serum concentrations >75 nmol/L) had more live births (odds ratio (OR): 1.33; 95% confidence interval (CI): 1.08 to 1.65), more positive pregnancy tests (OR: 1.34; 95% CI: 1.04 to 1.73) and more physiological courses of pregnancy (OR: 1.46; 95% CI: 1.05 to 2.02), compared to women with serum 25(OH)D3 deficiency or insufficiency [47]; however, no association between miscarriage and vitamin D status was found [47].

Again, a study of 1191 women with previous pregnancy losses showed that women with preconception levels of 25(OH)D3 ≥75 nmol/L were more likely to achieve pregnancy (adjusted RR: 1.10; 95% CI: 1.01 to 1.20) and give birth to a live newborn (RR: 1.15; 95% CI: 1.02 to 1.19) than those with lower serum levels (<75 nmol/L) [48].

Current evidence from clinical studies indicates that vitamin D deficiency is very common among women with CD and is associated with a variety of adverse outcomes, including reproductive ones [49]. Despite the promising results of certain studies [41,42], whether vitamin D supplementation represents a potential supportive therapeutic option for women with CD is controversial [43]. 

Furthermore, we are not aware of studies examining the potential role of vitamin D and fertility in women affected by CD. Therefore, to date, no final conclusion can be drawn regarding the potential beneficial clinical effects of vitamin D. The authors recommend further research including investigations on optimal vitamin D delivery to overcome malabsorption in CD and ideal vitamin D concentrations to maximize extra bone benefits, including fertility.

#### 2.1.3. Iron

Iron is the micronutrient that has been most studied in IBD, given the high prevalence of IBD patients who suffer from anemia [5], especially women who are at higher risk of developing iron deficiency than men. In IBD, the most common cause of iron deficiency is intestinal mucosa inflammation impact (i.e., increased loss due to blood loss from the gastrointestinal tract, and malabsorption) [50,51].

In patients with CD and iron deficiency, hepcidin plays a key role in regulating iron homeostasis (e.g., direct inhibitor of ferroportin) [50], being positively correlated with pro-inflammatory cytokines increase, including IL-6, IL-1, IL-17, and TNF-alpha [52].

Few studies have reported the association between serum iron concentration and ovulatory functions or women’s fertility [53]. A prospective cohort study by Chavarro J.E. et al. [54], involving 18,555 premenopausal women (US nurses aged 24 to 42 years at enrolment) without history of infertility, evaluated whether iron supplementation or intake of total heme and non-heme iron was associated with lower risk of ovulatory infertility. Women who were supplemented with iron had a significantly lower risk of ovulatory infertility than women who were not (RR: 0.60; 95% CI: 0.39–0.92). The same study also found an inverse association of non-heme and total iron consumption with infertility [54]. A recently published study aimed at ascertaining the effects of severe iron deficiency on fertility in female rats found that there was a significantly lower conception rate in the iron-deficient group compared to the controls [55]. Those results suggest that iron impact on female reproductive function begins before pregnancy. Pregnant women have a substantial increase in iron requirements [56], and maternal deficiency increases the risk of adverse birth outcomes and may cause irreversible adverse effects on neurodevelopmental outcomes in the offspring, as well as increase fetal–maternal morbidity and mortality [57]. 

#### 2.1.4. Zinc 

Zinc acts as a cofactor, and it is intimately involved in the regulation of both innate and adaptive immune response [58], thus may influence CD pathogenesis through several mechanisms. It plays a role in innate immunity through its effect on natural killer cells, macrophages, and neutrophils function and in adaptive immune responses by T- and B-lymphocyte functions [59]. Zinc downregulates NF-kB signaling indirectly, leading to a decreased expression of its downstream products (namely IL-6 and TNF-alf [58,60]. Zinc also exerts an anti-oxidative effect through its inhibition of NADPH oxidase and its role as a cofactor for superoxide dismutase [61].

Zinc deficiency has been described in a significant portion of IBD patients (15.2–65%) [62,63]. Losses are increased in association with chronic diarrhea, high output ostomies, and fistulas, which often pose a problem for IBD patients [6]. Those increased losses combined with the impaired absorption due to inflammation and as well as to medications (i.e., glucocorticoids) are likely responsible for zinc deficiency in patients [6].

A prospective study of 170,756 women from the Nurses Health Study I and Nurses Health Study II, followed for 26 years, found that increased zinc intake was inversely correlated with CD risk [64]. Furthermore, Siva S. et al. [65], in a cohort study of 773 CD patients (421 women), highlighted the relationship between zinc deficiency and adverse patient outcomes. They found that zinc deficiency (equivalent to serum zinc concentration <0.66 mcg/mL) was associated with a significant increase in hospitalizations (CD: OR 1.44.; 95% CI: 1.02–2.04), surgery (CD: OR 2.05; 95% CI: 1.38–3.05), and complications (CD: OR 1.50; 95% CI: 1.04–2.15) [65]. Furthermore, in patients who were able to achieve normal zinc levels within 12 months, the risk of adverse outcomes returned to physiological levels normal [65].

On the other hand, zinc has a pivotal role in the proper functioning of the reproductive system, regulating cell differentiation and proliferation [24]. It is required for ovulation, fertilization, normal pregnancy, fetal development, and parturition [24]. Despite this, less is known about the effects of zinc on female reproductive system and fertility, as only relatively few investigations have been performed [24,66]. The majority of studies focus on the role of zinc (and zinc supplementation) in pregnancy and fetal development.

The relationship between zinc and fertility in women with CD makes this nutrient worth further investigation in order to better highlight the potential impact of dietary zinc and zinc supplementation on the fertility of women affected by CD.

### 2.2. Fiber

Fiber has a key role on human gut; however, patients with CD often report an increase in symptoms after dietary fiber intake, which might be due to food and gut microbiota (GM) interactions, but is not yet fully understood [67,68].

Dietary fiber interacts with microbial populations, leading to different outcomes, including increased food viscosity and increased intestinal emptying time and nutrient release [69], thus reducing glycemic response [70]. Several studies have reported dysbiosis in patients affected by CD; however, a direct causal relationship between dysbiosis and IBD has not been established in humans [71,72,73,74]. Individual differences in GM composition, which predominantly collects four different phyla, Firmicutes (60–90%), Bacteroidetes (8–28%), Proteobacteria (0.1–8%) and Actinobacteria (2.5–5%) [75,76], result in different scenarios depending on microbial abundance. Microbes break down chemical bonds through very complex metabolic pathways and produce short chain fatty acids (SCFAs), essential for human health [77,78]. In particular, propionate, acetate, and butyrate are essential energy sources for colonocytes as well as for the maintenance of intestinal homeostasis through anti-inflammatory action [79,80]. Changes in fermentation processes can potentially interrupt the gut microenvironment, immune modulation, and energy metabolism [81,82]. 

Ananthakrishnan et al. in the Nurses’ Health Study, a prospective study in which 170.776 adult women were followed over 26 years [83], concluded that long-term fiber consumption was associated with a reduced risk of developing CD [83]. Compared with the lowest quintile of energy-adjusted cumulative average intake of dietary fiber, intake of the highest quintile (median of 24.3 g/day) was associated with a 40% reduction in risk of CD (multivariate HR for CD, 0.59; 95% CI 0.39–0.90) [83]. Is noteworthy that fiber consumption in the highest quintile is in agreement with the fiber intake recommended for the general population [84]. Among different food sources of fiber, only fiber from fruit, but not from other vegetables, cereals, or legumes, was associated with a reduced risk of developing CD [83]. 

In a systematic review, Wedlake et al. [85] concluded that none of the studies included in the review demonstrated a benefit on disease outcome, considering either remission or active phase [85]. Again, Hou JK et al. in their systematic review reported that a high-fiber diet was associated with a decreased risk of developing CD, but no evidence supports its role in the disease active phase treatment [86]. 

However, in a more recent cohort study, the authors [87] concluded that subjects in remission from CD (*n* = 1130) in the highest quartile of fiber consumption were less likely to have a flare-up phase (OR] 0.58, 95% CI 0.37–0.90). Women who did not avoid high-fiber foods during the 6 months of follow-up were 40% less likely to have a disease flare than those who avoided high-fiber foods (adjusted OR, 0.59; 95% CI, 0.43–0.81) [87]. 

Furthermore, a low-fiber diet is commonly associated with a high glycemic index diet, which increases insulin levels, leading to consequent insulin resistance and impaired oxidative status, threatening the delicate balance of ovarian function [88]. Moreover, a high concentration of insulin can also over-regulate the production of free testosterone, resulting in hyperandrogenism [89]. 

For instance, Chavarro et al. [90] found that dietetic glycemic index was positively related to ovulatory infertility among nulliparous women; the same was not observed among parous women (*p*, interaction = 0.02). The risk ratio (95% CI) related from the top to the bottom quintile of dietary glycemic index was 1.55 (1.02–2.37) (*p*, trend = 0.05) and 0.78 (0.51–1.18), respectively, among nulliparous women and parous women (*p*, trend = 0.22) [90].

In agreement, in a parallel web-based prospective preconception cohort in Denmark and North America, Willis et al. [88] observed that diets with high glycemic load (GL), high carbohydrate-to-fiber ratio, and added sugars were associated with modestly reduced fertility [88], while a diet with an adequate intake of fiber rather than simple sugars supports fertility by modulating insulin sensitivity and GM functions [88]. 

### 2.3. Prebiotics

Prebiotics are classified as typically indigestible fiber with functional properties that can selectively stimulate specific intestinal bacteria and produce SCFA [91]. Over the years, several small studies have been conducted to evaluate the efficacy of prebiotics in the treatment of CD, but data are divergent. In a small open-label trial with a sample of 10 patients with active ileocolonic CD, administered 15 g of inulin daily for 3 weeks [92], the authors observed a significant reduction in disease activity and increased concentrations of fecal bifidobacteria. However, those results were not confirmed by a well-powered placebo-controlled trial (*n* = 103), even though the same prebiotic was administered [93].

There is currently no convincing evidence in support of prebiotics for CD treatment. Despite the interest of the scientific community, no experimental studies present in the literature, nor meta-analyses, have so far been able to support those assumptions.

The role of prebiotics was also investigated in PolyCistyc Ovary Syndrome (PCOS), one of the most common endocrinological conditions in women with infertility [94], suggesting a key role in counteracting dysbiosis due to GM [95]. With their functions, prebiotics affect the composition of the GM, which may be involved in the pathogenesis of PCOS [96].

Fructoligosaccharides, inulin, galactooligosaccharides and lactulose are the most widely used prebiotics [97].

Several studies have shown that the prebiotic stimulation of Bifidobacterium and Lactobacillus growth provide a positive impact on metabolic markers and fasting blood glucose reduction, as well as serum triglycerides and LDL cholesterol decrease with HDL cholesterol increase [98]. Shamasbi et al. conducted a trial in which 20 g of resistant dextrin was administered to women with PCOS for 3 months [99], observing an improvement in glycemic and lipid profile. The same study reported that the consumption of dextrin improved the irregularity of the menstrual cycle as well as hirsutism and androgen serum levels [99]. 

However, further studies are needed to better evaluate the effect of prebiotics on the gut microbiota and the consequent effect on PCOS determinants.

GM has also begun to be investigated in women with endometriosis, another cause of infertility [100,101,102]; however, the studies conducted so far have not explored the role of prebiotics in endometriosis management. 

### 2.4. Probiotics

According to FAO/WHO, probiotics can be used to manipulate the microbiome plasticity and are considered a potential therapy for CD [103].

Several studies, unfortunately with small size samples, have been conducted to explore the role of probiotics in relation to IBD. An open-label, uncontrolled study conducted on 10 patients with active disease who were given a probiotic containing *Bifidobacterium breve, Bifidobacterium longum, Lactobacillus casei* and a prebiotic (psyllium) reported symptom improvement in seven patients, and two of them stopped the corticosteroid therapy [104]. In another randomized double-blind placebo study, 35 patients with active disease underwent therapy with *Bifidobacterium longum* and Synergy 1 [105]. After 6-month follow-up, the results revealed that 62% (8 out of 13) and 45% of the subjects (5 out of 11) had symptom improvement, respectively, in the treated and the control group. Despite drop-out, the positive outcome registered in most treated subjects suggested this probiotic mix as a good treatment in active CD in clinical practice [105].

Other studies reported interesting results with a probiotic blend, VSL#3, containing 900 billion freeze-dried bacteria belonging to four strains. Fedorak et al. used this mixture to evaluate the endoscopic relapse rates, 90 and 365 days post-CD surgery. Although the differences in endoscopic recurrence were not significant between the two groups, a lower level of pro-inflammatory cytokines in the intestinal mucosa was noted in treated patients [106].

Bourreille et al. [107] administered yeast *Saccharomyces boulardii* to 165 subjects for 52 weeks during the remission phase after steroids or salicylates therapy, but no significant differences were observed between the treatment group compared to the placebo group [107].

Definitively, Cochrane reviews found insufficient evidence for the effectiveness of probiotics in patients with CD as for remission induction and maintenance, or the prevention of postoperative recurrence [108].

Concerning probiotics and fertility, GM may contribute to regulate endocrine and metabolic disorders, including PCOS, and thus fertility [109]. Several studies reported an improvement in glycemic and lipid profile after probiotics treatment. Rashad et al. [110] found that supplementation with *L. delbruekii* and *L. fermentum* for 12 weeks significantly reduced HOMA-IR and improved the lipid profile [110]. Positive results on the regulation of glucose and insulin levels were reported by Ahmadi et al. [111] after supplementation with *L. casei*, *L. acidophilus*, *L. rhamnosus*, *L. bulgaricus*, *B. breve*, *B. longum,* and *Streptococcus thermophilus* [111]. Heshmati et al. [112] conducted a meta-analysis of seven RCT and found a significant effect on glucose and insulin levels, also reporting an increase in HDL levels and a decrease in triglyceride serum levels. However, they found no significant effect in anthropometric indices (weight, BMI and waist circumferences), HOMA-IR and LDL levels between treated group of PCOS women and controls [112]. 

Again, several studies have investigated the role of probiotics on GM modulation and fertility in patients with CD. Unfortunately, the results obtained were often not significant and the sample size too small. Many more studies are needed to prove the role of probiotics on women’s fertility.

### 2.5. Polyunsaturated Fatty Acid

Polyunsaturated fatty acids Omega-3 (ω-3) and Omega-6 (ω-6) are thought to have different effects on inflammation [113]. Conventionally, ω-6 s are considered to be pro-inflammatory [113]; on the contrary, ω-3 s are important for inflammation suppression [113].

Omega-6 polyunsaturated fatty acids include linoleic acid, convertible into arachidonic acid, which, in turn, represents a precursor of inflammatory mediators such as prostaglandins and leukotrienes [114]. On the other hand, α-linolenic acid, an essential ω-3 PUFA, is a precursor of eicosapentaenoic acid (EPA) and docosahexaenoic acid (DHA), with anti-inflammatory properties [114]. EPA and DHA are involved in the regulation of immunological and inflammatory responses by inhibiting inflammatory process genes [115]. 

ω-3 and ω-6 imbalance is regarded as a critical epigenetic factor in the pathogenesis of several diseases, including CD [113]. Moreover, recent evidence suggests that an imbalance of PUFA metabolism also plays a pivotal role in different clinical conditions, including infertility [116]. Indeed, ω-3 can positively impact fertility, as it plays an essential role in steroidogenesis [117] and, as already mentioned, has significant anti-inflammatory properties [113]. 

The Mumford et al. [118] study, evaluating the associations between total and specific types of dietary fat intake (i.e., MUFAs, SFAs, PUFAs, specifically, a-linolenic acid, EPA, DHA, and trans fats) and the risk of sporadic anovulation in regularly menstruating women (*n* = 259; aged 18–44 years) in the BioCycle Study [119], demonstrated that an increase in DHA consumption from oily fishes was associated with a lower risk of anovulation. Again, Hammiche F. et al. [120] demonstrated that ω-3 (i.e., ALA, EPA, and DHA) significantly contributes to the number of follicles and embryo morphology in women (*n* = 235; mean age 35 years) undergoing in vitro fertilization (IVF)/intracytoplasmic sperm injection (ICSI) treatment. 

Overall, the results of the association between ω-3 and fertility are contradictory. Indeed, according to Stanhiser et al. [121], no association was observed between serum concentration of ω-3 and the probability of improving natural fertility in women (*n* = 200, aged 30–44 years).

Similarly, despite experimental evidence showing biological and physiological plausibility, clinical data on the benefits of ω-3 are still debatable and conflicting in CD [122,123]. Clinical trials have produced mixed results, showing beneficial effects (e.g., reduction of inflammation and decreased need for steroid therapy), but at the same time failing to demonstrate a clear protective effect in preventing clinical relapse [115,124]. Lev-Tzion et al. [122] systematically reviewed ω-3 efficacy in maintaining CD remission. The primary outcome was relapse rate during the observation time, and six studies (total sample *n* = 1039) were included. Evidence from two large, high-quality studies (EPIC studies) [125] suggested that ω-3 is probably ineffective in CD remission maintenance.

Despite the EPIC studies’ results [125], in our opinion, it is inappropriate to dismiss the potential of ω-3 to prevent relapse in CD. Indeed, Turner et al. [123], in a Cochrane systematic review of a significant sample (*n* = 1039 individuals with CD), did not rule out strong evidence of any beneficial effect of ω-3. Moreover, due to the potential effects of ω-3 on female reproduction and fertility [118,119,120], the consumption of foods rich in ω-3 should nevertheless be recommended.

### 2.6. Conditionally Essential Amino Acids (Glutamine, Arginine)

Amino acids are the building blocks of protein synthesis and are involved in many other functions (e.g., gene expression, cell signaling, protein turnover, oxidative stress, and immunity) [126]. A systemic inflammatory condition may cause malnutrition and general glutamine deprivation, which are associated with loss of muscle tone, fatigue, and depression [127]. Glutamine is a conditionally essential amino acid, since when the catabolic stress increases, requirement become essential [128]. Glutamine is the main substrate for enterocytes and immune cells [129]. Moreover, its role is linked to intestinal permeability regulation, maintaining the integrity of tight junctions [130]. Conversely, glutamine deprivation or glutamine synthase inhibition leads to an increase in paracellular permeability and a decreased expression of tight junction proteins [131]. In the colon tissue of IBD patients in remission, glutamine levels are reduced compared with controls. This difference is even greater during active illness [132]. Although glutamine supplementation has been shown to be useful in murine models, the same does not occur in CD patients. From a meta-analysis investigating the efficacy and safety of glutamine supplementation for inducing remission in CD [133], two small RCTs, one of them on a pediatric population (*n* = 42), were analyzed [133]. The results did not report statistically significant changes in intestinal permeability between patients who received glutamine supplementation and those who did not [134]. 

Concerning arginine, which is also involved in gut systemic inflammation [135] and modulates inflammation through different biochemical pathways, it has been shown that micro vessels in CD can be characterized by endothelial dysfunction with loss of NO-dependent dilatation, thus contributing to the perpetuation of the inflammatory state [136]. Arginine reduces proinflammatory cytokines throughout NO pathway immunomodulation, with no inflammatory response exacerbation [136,137]. However, studies on CD patients are limited, small in size [138], and report controversial results [134,139]. 

Even though arginine and glutamine have aroused the interest of the scientific community starting from the results of Battaglia et al. [140], the studies so far have not been able to demonstrate a relationship between the over mentioned amino acids and fertility in women with CD; therefore, further research is needed. 

## 3. Conclusions

CD is a complex disorder involving immune dysregulation, leading to infiltration and destruction of the gastrointestinal tract. Sex differences in incidence and prevalence have been reported in CD, and there are gender-specific issues that physicians need to recognize, including the impact on fertility in women of childbearing age. There is no doubt that infertility increases during the active phase of the disease, and those women of childbearing age, in particular, need to undergo a cautious and far-sighted nutritional assessment since diet and lifestyle seem to be significant factors influencing fertility. Furthermore, serum concentrations of micronutrients and vitamins should be monitored, and in the case of deficiencies, supplementation should be initiated.

Disease control is undoubtedly key to ensure women with CD have careful and accurate reproductive planning and personnel wellbeing. A good state of health can be achieved by expanding knowledge about the pivotal role of nutrition in controlling CD impairment of nutritional status and supporting women’s fertility (as indicated in Table 1). 

Multidisciplinary approaches to nutritional care are increasingly emphasized and recommended acknowledging the interrelationship role between different professionals (i.e., dietitians, health coaches, clinical nutritionists, gastroenterologists, gynecologists, psychologists) and their complementary contributions towards the delivery of optimal food and nutritional care throughout continuous personalized care.

## Figures and Tables

**Figure 1 nutrients-14-02423-f001:**
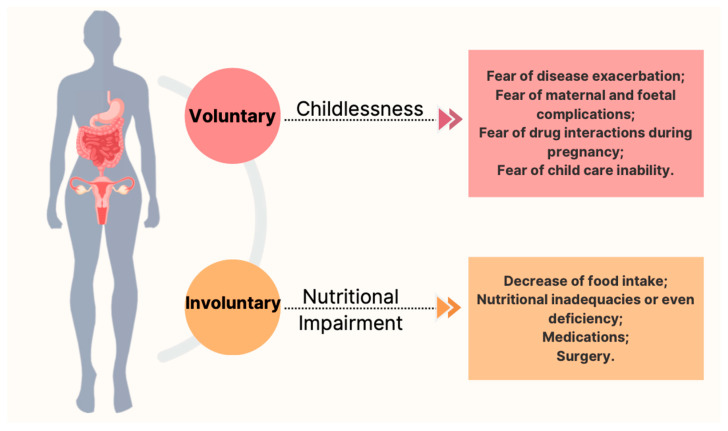
The most common causes of voluntarily and involuntarily infertility in women with CD.

**Table 1 nutrients-14-02423-t001:** The role of nutrients in CD and women fertility.

Nutrient	Role		Risk Factors for Deficiency/Inadequate Intake
	CD	Fertility	
**Vitamin D**	- regulation of immunity and immune response (gut mucosal immunity and integrity) [23,34]; - inhibiting the production of proinflammatory cytokines [35].	- modulation of female reproductive and fertility, as well as pregnancy and lactation [44,45,46]; - vitamin D receptors expression in numerous tissues of the reproductive organs (e.g., ovaries, endometrium, placenta, pituitary gland, hypothalamus) [26,46]; - steroidogenesis [46].	- intestinal inflammation leading to impaired absorption of nutrients [37]; - bile acid malabsorption [37]; - restricted dietary intake [37]; - reduced sunlight exposure [37]; - medication (immunosuppressive treatment) [37].
**Fiber**	- modulation in gut microenvironment through interaction with microbiota (maintenance of intestinal homeostasis, anti-inflammatory action, production of SCFA) [77,78,79,80,81,82].	- modulation in glycemic control [88]; - modulating insulin sensitivity [88].	- low dietary intake (e.g., low consumption of fiber-containing foods) [68].
**Folate**	- prevention of HHcy and related consequences (e.g., oxidative stress, cardiovascular diseases and arterial and venous thromboembolic events) [29].	- modulation of fertility and reproductive process (gametogenesis and early embryogenesis) through prevention of HHcy [32].	- low dietary intake (e.g., low consumption of folate-containing foods including green leafy vegetables, fruits, bread, and fortified cereals) [5]; - malabsorption [5]; - medications (e.g., acting through inhibition of dihydrofolate reductase (methotrexate) or favoring malabsorption (sulphasalazine)) [5].
**Vitamin B12**			- malabsorption due to resection of more than 60 cm of terminal ileal [5].
**Iron**	- modulation of inflammation through the increase in hepcidin (positively correlated with the increase in pro-inflammatory cytokines, such as IL-6, IL-1, IL-17, and TNF-alpha) [52].	- lower risk of ovulatory infertility [90]; - decrease risk of adverse birth outcomes [57].	- increased inflammation of the intestinal mucosa (i.e., its increased loss due to blood loss from the gastrointestinal tract, and malabsorption) [50,51].
**Zinc**	- regulation of both the innate (affecting the function of natural killer cells, macrophages, and neutrophils) and adaptive arms (influencing the function of T- and B-lymphocytes) of the immune system [58,59]; - anti-oxidative effect (through inhibition of NADPH oxidase, and its role as a cofactor for superoxide dismutase [61].	- regulation of cells differentiation and proliferation [24].	- Increase losses (chronic diarrhea, high-output ostomies, and fistulas [6]; - medications (reduced absorption (glucocorticoids)) [6].
**Omega-3**	- regulation of immunological and inflammatory responses [115]; - reduction of inflammation [115].	- anti-inflammatory properties in steroidogenesis [113,117].	- malabsorption [10].
**Arginine**	- intestinal permeability regulation (maintenance of the integrity of tight junctions) [130].	- the collected studies do not demonstrate a relationship between the role of glutamine and arginine in CD, related to fertility.	- misregulation of inflammatory processes [136,137].
**Glutamine**	- modulation of the gut inflammation (reducing pro-inflammatory cytokines [135].		- increase in paracellular permeability [131].

SCFA: short-chain fatty acid.

## Data Availability

Not applicable.

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
