# Peer review of "Crohn’s Disease and Female Infertility: Can Nutrition Play a Supporting Role?"

_nutrients, 2022, doi:10.3390/nu14122423_

Round 1

Reviewer 1 Report

 In my opinion, English language is understandable, and the work does not require any language editing. Authors review very interesting topic about the impact of nutrition on female infertility in Crohn’s disease patients.

Below I present what should be improved at this manuscript:

Please correct the chapter numbering, there is 1. Introduction, 2. CD and fertility: the supporting role of nutrition and 4. Conclusions. Chapter number 3 is missing. There is also incorrect numbering in the subsections – after 2.1.4. Zinc there is 3.2. Fiber.

Add a Figure- a proposal 1) the most common causes of infertility (childlessness) in women with CD, or 2) micronutrient deficiencies induce infertility throughout different mechanisms- ovulation, implantation, etc.).

In Table 1 there is an information about Arginine and Glutamine, that "the collected studies do not show a relationship between the role of glutamine and arginine and fertility". Below I present a few works describing this influence. If the Authors meant that there is no such evidence in patients with CD, please include it in the description, otherwise the Reader may be misled while analyzing Table 1.

-        Arginine, glutamine, tryptophan and taurine play a crucial role in fetal growth, development and survival (…) [Hussain T, Tan B, Murtaza G, Metwally E, Yang H, Kalhoro MS, Kalhoro DH, Chughtai MI, Yin Y. Role of Dietary Amino Acids and Nutrient Sensing System in Pregnancy Associated Disorders. Front Pharmacol. 2020 Dec 22;11:586979. doi: 10.3389/fphar.2020.586979.]

-        These nutrients include select amino acids in histotroph (arginine, leucine and glutamine of particular interest) that stimulate conceptus growth and development, as well as interactions between maternal uterus and the conceptus, thus impacting maintenance of pregnancy, placental growth, development and functions, fetal growth and development, and consequential pregnancy outcomes [Gao H. Amino Acids in Reproductive Nutrition and Health. Adv Exp Med Biol. 2020;1265:111-131. doi: 10.1007/978-3-030-45328-2_7].

-        There is evidence that amino acids such as leucine, arginine and glutamine could stimulate the mammalian target of rapamycin (mTOR) pathway, which plays a pivotal role in primordial follicle activation [Alborzi P, Jafari Atrabi M, Akbarinejad V, Khanbabaei R, Fathi R. Incorporation of arginine, glutamine or leucine in culture medium accelerates in vitro activation of primordial follicles in 1-day-old mouse ovary. Zygote. 2020 Jun 2:1-8. doi: 10.1017/S096719942000026X].

-        Dietary supplementation of amino acids during pregnancy could help mitigate reproductive disorders and thereby improving fertility in animals as well as humans [Hussain T, Tan B, Murtaza G, Metwally E, Yang H, Kalhoro MS, Kalhoro DH, Chughtai MI, Yin Y. Role of Dietary Amino Acids and Nutrient Sensing System in Pregnancy Associated Disorders. Front Pharmacol. 2020 Dec 22;11:586979. doi: 10.3389/fphar.2020.586979].

This recommendations only refines the overall draft of the manuscript. The manuscript submitted by the Authors is in line with the subject of the Nutrients, and will be an attractive article for the Readers.

Reviewer 2 Report

Dear Authors,

The manuscript (nutrients-1763331) submitted for review is very interesting. 

Authors, Please note and address the following comments:

Introduction

The Introduction section is well written. In my opinion, at the end of the Introduction section, the purpose of this Review should be added.

Material and Methods:

It is a pity that the authors did not write about the methods they used to search for articles for review. In which databases were the articles searched? What was the key to choosing these references?

Conclusions

What are the practical and theoretical implications of the research? What are the authors' recommendations for scientists?

Conclusions should not contain references. The present conclusions are not the conclusion. In my opinion, it should be re-written.

References

Many references (76 out of 139) come from the last 7 years, 33 from 2010-2014, 22 from 2000-2009, and 9 from before 2000. References are cited according to the journal’s regulations

 Despite my comments, I am pleased to recommend this manuscript for publication. I believe that it concerns an important area of research in an international context.

 Reviewer
